# Investigation of the Temperature Actions of Bridge Cables Based on Long-Term Measurement and the Gradient Boosted Regression Trees Method

**DOI:** 10.3390/s23125675

**Published:** 2023-06-17

**Authors:** Fen Wang, Gonglian Dai, Yonglu Liu, Hao Ge, Huiming Rao

**Affiliations:** 1School of Civil Engineering, Central South University, Changsha 410075, China; wfone1@csu.edu.cn (F.W.);; 2Hunan Provincial Key Laboratory of Power Electronics Equipment and Gird, Changsha 410083, China; 3School of Automation, Central South University, Changsha 410083, China; 4Southeast Coastal Railway Fujian Co., Ltd., Fuzhou 350013, China

**Keywords:** cable-stayed bridges, design representative value, extreme value analysis, GBRT method, long-term temperature test, temperature field

## Abstract

Cable-stayed bridges have been commonly used on high-speed railways. The design, construction, and maintenance of cable-stayed bridges necessitate an accurate assessment of the cable temperature field. However, the temperature fields of cables have not been well established. Therefore, this research aims to investigate the distribution of the temperature field, the time variability of temperatures, and the representative value of temperature actions in stayed cables. A cable segment experiment, spanning over one year, is conducted near the bridge site. Based on the monitoring temperatures and meteorological data, the distribution of the temperature field is studied, and the time variability of cable temperatures is investigated. The findings show that the temperature distribution is generally uniform along the cross-section without a significant temperature gradient, while the amplitudes of the annual cycle variation and daily cycle variation in temperatures are significant. To accurately determine the temperature deformation of a cable, it is necessary to consider both the daily temperature fluctuations and the annual cycle of uniform temperatures. Then, using the gradient boosted regression trees method, the relationship between the cable temperature and multiple environmental variables is explored, and representative cable uniform temperatures for design are obtained by the extreme value analysis. The presented data and results provide a good basis for the operation and maintenance of in-service long-span cable-stayed bridges.

## 1. Introduction

With the rapid development of sea-crossing bridges on high-speed railways (HSRs), the rate of constructing cable-stayed bridges has increased significantly in the past decade [1,2,3]. For cable-stayed bridges, cables are the primary load-carrying members. Due to long-term exposure to the atmospheric environment, bridge cables are constantly subjected to daily and seasonal fluctuations of environmental variables such as air temperature, solar radiation, and wind [4,5]. These environmental variables contribute to the change in temperature fields or thermal actions. With the increasing span and the frequent occurrence of extreme weather events, thermal actions on bridge cables become more severe, presenting challenges to bridge design, construction, and maintenance [6]. Cables are affected by thermal actions in two different ways. One is to cause the longitudinal expansion and contraction of steel wires due to uniform temperature changes [7]. The other is to induce temperature gradients along cable cross-sections. Additionally, these thermal actions can cause secondary internal forces in the cables that will jeopardize the stability of the bridge. Therefore, clarification of temperature action patterns is crucial for accurately determining the thermal effects on cable-stayed bridges.

Researchers have paid close attention to the thermal action that is regarded as a significant external loading of bridges. The majority of published studies have mainly focused on the thermal actions of concrete towers [8], concrete box girders [9], steel girders [10], and composite girders [11,12]. For design purposes, Lou et al. [13] and Liu et al. [14] used a statistical analysis method and simplified temperature fields to clarify the uniform temperature variation pattern of box girders, the temperature gradient pattern, and the calculation of temperature extremes. These available methods are generally applicable to the temperature study of large components such as bridge girders and piers [15,16] but are not suitable for small components such as cables, which are more sensitive to temperature changes [17]. As for stayed cables, most of the reported studies focused on thermal effects on cable forces [18,19,20] and the damage detection of cable structures [21]. Some researchers have investigated the thermal actions of stayed cables, mainly by using the finite element method and short-term (a few days or weeks) field experiments [22,23]. However, in a short-term test, the temperature changes little and is constrained within a relatively narrow range. Consequently, the derived temperature distribution patterns are deemed incomplete or not representative.

Realistic temperature data can be recorded by positioning the temperature sensors on a real bridge cable or a segment model. Based on one-month tests on a bridge cable, Zhang et al. [24] found that the length change was mainly due to the change in the cable uniform temperatures. The uniform temperature during the design life of the cable is generally represented by a representative cable temperature. For design purposes, a representative value of each temperature action is preferred, which needs to be determined [25]. However, the temperatures obtained from a short-term test are not representative of the entire bridge life. An inaccurate prediction of representative temperatures will result in significant erroneous calculations of thermal deformations, especially for long-span stayed-bridge bridges [26]. For this reason, long-term temperature tests are required to provide sufficient data for statistical analysis of temperature fields.

During the design and construction processes of cable-stayed bridges, it is critical to determine the representative temperatures of cables. In the Chinese bridge design specifications [27], the stipulated temperature pattern for cables is deemed approximate, for which only the temperature difference between the cable and the girder (or tower) is given and the temperature pattern and representative temperature are not specified. A similar issue also exists in the Euro codes [28]. The AASHTO standard [29] specifies the maximum and minimum uniform temperatures for steel structures based on various meteorological conditions. The Australian codes [30] divide the nation into a dozen zones and determine the temperatures of cables based on the relationship between the air temperature and the structural temperature in each zone. From the existing design codes, there are no clear regulations on temperature distribution patterns and representative temperatures of stayed cables, which hinders the design and construction of cable-stayed bridges. In addition, the thermal effects on cables are highly time-dependent and spatial. However, the variation in temperatures (short- or long-term) is not mentioned in any existing code. In the analysis of representative temperatures, annual extreme values are considered ideal samples. However, in engineering practice, there are rarely long-term temperature monitoring projects that exceed 10 years. This makes it challenging to obtain a sufficient sample size for extreme value analysis. With limited data, predicting representative temperatures for the entire service life of a structure becomes a challenging task. Therefore, with the widespread application of machine learning algorithms [31,32,33], it is possible to introduce machine learning methods in structural temperature research to establish the relationship between structural temperature and environmental variables. This can be used to calculate long-term structural temperature data and supplement the sample size. It is of great significance for predicting representative temperatures with different return periods.

This paper aims to investigate the distribution of temperature fields, the time variability of temperatures, and the representative values of temperature actions in stayed cables. An experimental cable segment is fabricated and instrumented for a temperature test lasting more than one year. Machine learning is adopted to establish the relationship between the cable temperature and multiple environmental variables. A more complete approach is proposed to establish the temperature distribution patterns and time variability of cables, resulting in the derivation of a long-term trend variation curve and a short-term fluctuation variation curve, and the generation of the representative cable temperatures with a 50-/100-year return period by extreme value analysis (EVA). Finally, the presented temperature field and representative temperatures provide a good basis for the operation and maintenance of in-service long-span cable-stayed bridges.

## 2. Research Background

### 2.1. Description of the Bridge and Segment Test 

The experimental cable segment is taken from a long-span cross-sea cable-stayed bridge in China, named “the Quanzhou Bay Cross-Sea High-Speed Railway Bridge (QBCB)”, which is located at the longitude 118°42′ east and the latitude 24°50′ north. The bridge spans over Quanzhou Bay in the south–north direction and connects to Quanzhou City along the same direction. As shown in Figure 1, the main bridge is a double-tower, double-plane cable-stayed bridge with a main span of 400 m. The span layout is 70 m + 130 m + 400 m + 130 m + 70 m, totaling 800 m. The inclination angle between the longitudinal axis of the main girder and the north direction is 38°. The towers are H-shaped concrete towers with a height of 160.254 m. There are 18 cables on each side of the bridge tower, and the angle between the cables and the tower changes from 61° on the innermost side to 13° on the outermost side. The stayed cable adopts a spatial double-plane system with a fan-shaped arrangement and the cable spacing on the main girder is 10.5 m. The longest cable is about 218.2 m, and the main specifications are PES(C)7-211 and PES(C)7-151. The cables are composed of parallel 7 mm-diameter steel wire bundles with a strength of 1670 MPa, and the outer sheath is made of high-density polyethylene (HDPE).

Stayed cables are the primary load-bearing components and play an important role in cable-stayed bridges, which are considerably influenced by temperatures. During prolonged exposure to the atmospheric environment, cables are directly subjected to daily and seasonal fluctuations of air temperatures and solar radiations as well as irregular winds. These environmental factors cause changes in the temperature fields of cables (i.e., thermal action) and the cable forces affecting the overall force state of a cable-stayed bridge. Therefore, defining appropriate temperature patterns is an important prerequisite for accurately evaluating the thermal effects on stayed cables. The analysis of the internal temperature distribution of cables can provide a basis for studying the variation in the cable force under the influence of long-term environmental factors. The temperature test for the experimental cable segment is carried out near the bridge site for more than one year.

### 2.2. Experiment Process

In practice, it is difficult to arrange the temperature measuring points along a stayed cable, which may be several hundreds of meters long. The cross-sectional area of a stayed cable is generally small, and the main material is steel wire, which has a fast heat transfer rate. Therefore, for a single-stayed cable, the temperature variation along its length is usually not considered and the influence of the temperatures along the longitudinal direction is ignored. Thus, the proposed measurement scheme uses the cable segment test to reflect the overall temperature effect of stayed cables.

The test cable segment adopts the same technical specifications as in the QBCB. The segment length is 2 m, based on the specification of PES(C)7-151. The cable is composed of 151 parallel steel wire bundles encased under a white HDPE sheath. The outer diameter of the cable is 113 mm, and the thickness of the HDPE sheath is 9 mm.

The cross-section at 1/2 length is taken as the measurement section and 5 temperature sensors are installed on this section. After the factory prefabrication is completed, the steel wire at the measuring point is pulled out to half its length. The two ends of the cable are packaged, waterproofed, and transported to the test site. The PT100 temperature sensor is set on the measurement section and finally, the cross-section at both ends of the cable segment is encapsulated with thermal insulation adhesives. The test site is near the bridge site; there is no shading throughout the day, which is the typical environment of the bridge. The cable segment is placed horizontally and aligned with the main bridge, with the longitudinal axis oriented 38° east of north.

Relevant thermal phenomena of cables include heat conduction, solar radiation, and convective heat transfer (cooling). The heat transfer in the cable is mainly heat conduction. The form of heat transfer between the cable and the environment can be divided into solar radiation and convective heat transfer. For example, the wind speed in the environment affects the convective heat transfer between the cable and the air and the solar radiation mainly comes from the sun. The heat transfer and the temperature test scheme on the cable segment are illustrated in Figure 2a.

The temperature-monitoring system of the segment includes the temperature sensor, data acquisition instrument, and remote data transmission system, as shown in Figure 2b. A resistive temperature sensor PT100 with a temperature range of 40 to 80 °C and an accuracy of +0.1 °C is used in this test. The temperature sensor has a sample interval of 0.5 h and collects 48 data points per day (24 h × 2 data points/h). The ZY-108 data acquisition instrument is used in the cable temperature monitoring system. Each instrument has eight temperature measuring points with built-in memory and a sampling frequency of 10 Hz. The acquisition instrument uses the communication module ZY-34G to connect with the server through a 4G network. A remote data acquisition and control system is implemented to monitor and record cable temperatures.

### 2.3. Sensor Arrangement

The temperature sensors are arranged in the test section and divided into two groups. One group has one temperature sensor positioned in the center, which is 56.5 mm away from the outer edge of the cable. Another group has four temperature sensors positioned on the outer layer, which is 28.5 mm away from the outer edge. The cable is composed of parallel steel wires with a diameter of 7 mm, and it is covered by an outer HDPE sheath with a thickness of 9 mm. The locations of all temperature sensors are shown in Figure 3.

## 3. The Cross-Sectional Distribution and the Time-History Curve of Cables

### 3.1. The Cross-Sectional Distribution of Cables

The cross-sectional distribution and the time-history curve of structural temperatures are the two focal points of the study on the structural temperature fields. Analyzing the temperature distribution on the structure’s cross-section helps determine the representative structural temperatures, and investigating the time-history pattern of structural temperatures helps understand the trend between the temperature and the time for the most unfavorable situation. Based on the cable segment test of Quanzhou Bay, the above two points are studied. The temperature data of the cable segment test from September 2020 to September 2021 are shown in Figure 4.

The analysis of the data collected at each temperature measurement point shows that the amplitude of the temperature variation at each point is basically consistent. The measured point temperatures exhibit obvious seasonal variation characteristics in sine-curve form. Since the cable is a small member with excellent heat transfer performance [34], the temperature gradient within the cable cross-section is negligible. This is shown through an analysis of the experimental data.

To study the temperature distribution pattern of the cable, the cable cross-section is divided into inner and outer layers according to the location of the temperature sensors installed in the test. If there is no cross-sectional temperature gradient in the cable, the temperature of the inner layer measuring point (Sensor 3) should be consistent with that of the outer layer measuring point. Therefore, the temperature of Sensor 3 is used as the benchmark value, and Sensor 3′s temperature is subtracted from the temperatures of the other four measuring points to compare the temperature differences. The results are shown in Figure 5.

As seen in Figure 5, the maximum temperature difference between the four measuring points on the outer layer and Sensor 3 is basically between −1 and ~1 °C. The measured temperatures are then statistically analyzed to obtain the temperature’s mean and variance, as shown in Figure 6.

As seen in Figure 6, the temperature data collected at the five measuring points are generally consistent in terms of the mean and variance. Using measuring point Sensor 3 as the reference, the difference in mean or variance from any other measuring point is less than 5%, which is within the practically acceptable error range. The above analysis indicates that the temperature distributions on the cable cross-section differ little. In other words, the temperature distributions are generally uniform. A weighted average of the temperatures obtained from the five measuring points is used to represent the uniform temperature distribution over the cable cross-section. The uniform temperature (*T_u_*) on a cable cross-section is calculated by:(1)$$T_{u}=\sum_{i=1}^{5}\frac{A_{i}}{A}T_{i}$$
where *T_i_* is the temperature of sensor, *i*; *A_i_* is the area surrounding sensor, *i*; and *A* is the cross-sectional area of the cable. The cross-section is divided into five parts according to the arrangement of measuring points, as shown in Figure 7.

Therefore, the internal temperature field of the cable is deemed to be uniform when exploring the temperature variation within the cable and the relationship between the cable temperature and the environmental variables. That is, the uniform temperature variation along the cable cross-section can be used to describe the variation in the overall temperature of the cable.

### 3.2. The Time-History Curve of Cable Temperature

The observed temperatures vary with time with similar characteristics that are specified in Eurocode 1 [28]. According to long-term observation, the long-term trend variation and short-term fluctuation variation represent the majority of cable temperature variations. The long-term trend variation implies the annual temperatures of the cable, and it reflects the overall trend of the structure temperature time history (e.g., seasonable change in structure-atmospheric temperatures). The daily uniform temperature on the cable cross-section, *T_u_*, is calculated using Equation (1), and the daily average uniform temperature, *T_u,d_*, is used to define the average uniform temperature of the cable at any moment of the day. The annual temperature action is then represented by the variation in *T_u,d_*. The difference between *T_u_* and *T_u,d_* is defined as the daily uniform temperature fluctuation, *T_f_*, which represents the structure’s daily cycle of rising and falling temperatures induced by the influence of solar radiation. On the time-history curve of *T_u_*, *T_f_* appears as a “burr” on the temperature time-history curve with regular fluctuation, reflecting the short-term fluctuation of structural temperatures. *T_u,d_* and *T_f_* can be calculated by Equations (2) and (3), respectively.
(2)$$T_{u,d}=\sum_{i=1}^{n}\frac{T_{u,i}}{n}$$
where *T_u,i_* is the observed uniform temperature of the cable at time, *i*, of the day.
(3)$$T_{f}=T_{u,i}-T_{u,d}$$

### 3.3. Annual Cycle Variation in Uniform Temperatures on a Cable Section

In the research on long-term structure temperatures, the long-term variation trend is the seasonable variation in structure temperatures or the air temperatures, which may direct the change in structure temperatures. The long-term temperature trend of cables shows a significant regular variation with an annual cycle and the variation pattern of *T_u,d_* can be used to describe this long-term variation trend. The *T_u_*, *T_u,d_*, and *T_f_* of the cable during the one-year test period are calculated and plotted in Figure 8.

As seen in Figure 8, the *T_u,d_* curve shows seasonal variations, which are low in the winter and high in the summer, similar to the Fourier series. The *T_u,d_* time-history curve can be fitted as an extended formula of the Fourier series as:(4)$$T_{u,d}(t)=a_{0}-b_{0}\cos\left(\frac{2\pi }{365}\left(t-c_{0}\right)\right)$$
where *a*_0_ is the annual average uniform temperature of the cable (*T_u_*), *b*_0_ is the annual temperature variation amplitude, *t* is the daily sequence number starting from 1 January, and *c*_0_ is the initial phase. Based on the calculations, *a*_0_ = 22.25, *b*_0_ = 8.04, *c*_0_ = 22, and fitting accuracy *R*^2^ = 0.952. The fitting results are shown in Figure 8, which is expressed by
(5)$$\begin{array}{l} T_{u,d}(t)=22.25-8.04\cos\left(\frac{2\pi }{365}\left(t-22\right)\right)t\geq 0\\ \;R^{2}=0.952 \end{array}$$

According to the fitted results, the annual average uniform temperature of the cable in Quanzhou is 22.25 °C and the annual temperature variation amplitude is 16.08 °C. As seen in Figure 8, the uniform temperature of the cable varies as a sinusoidal function in an annual cycle. As a result of this temperature effect, the cable deforms due to the seasonal variation in expansions and contractions. The temperature actions can be divided into uniform heating action and uniform cooling action. Usually, the maximum heating and cooling actions occur in hot and cold seasons, respectively. Consequently, the collected temperature data may be split between the hot and cold seasons. The hot season runs from May to October and the remaining months (November to April) belong to the cold season. During the monitoring period of one year, the cable experienced a maximum *T_u_* of 38.07 °C and a minimum *T_u_* of 3.59 °C, giving the amplitude of *T_u_* a variation of approximately 34 °C. The highest *T_u,d_* is 33.59 °C (on 23 July) and the lowest value is 5.77 °C (on 22 January). Additionally, Figure 8 demonstrates that throughout the year, the cable fluctuation temperature, *T_f_*, varies more noticeably.

### 3.4. Daily Cycle Variation in Uniform Temperatures of Cable Section

#### 3.4.1. Statistical Analysis of Daily Cycle Variation

Short-term fluctuation variation refers to the fluctuation of daily temperatures caused mainly by the solar radiation intensity (SRI) which appears as a “burr” on the temperature time-history curve. The cable fluctuation temperature, *T_f_*, presents a daily cycle variation. A statistical analysis of *T_f_* is conducted. The time-history curve of *T_f_* obtained from the test is shown in Figure 9a. As seen, the time-history curve shows a periodical fluctuation during the test. Additionally, the fluctuation of *T_f_* is apparent despite the season (hot or cold).

To analyze the variation pattern of *T_f_*, two typical weather conditions, namely sunny and cloudy days, are selected for comparison. As shown in Figure 8, the difference between the measured air temperature and the cable *T_u_* for the two selected weather conditions is quite small, whereas the amplitude of the variation in fluctuation temperatures, *T_f_*, is obviously dissimilar between the two selected weather conditions. Figure 9b,c clearly show that *T_f_* has a significant positive correlation with SRI and that the variation in cable *T_f_* values has a certain time lag with the SRI. For the two selected weather conditions, the minimum and maximum *T_f_* values occur during the hours of 5:00–7:00 and 13:00–15:00, respectively. Clearly, the weather condition is a main factor affecting the SRI in the same region. Under sunny weather (no clouds), the weakening of solar radiation is not visible, and strong solar radiation will reach the ground. Under cloudy weather, the opposite is true. Clearly, sunny weather displays more apparent *T_f_* fluctuation, while cloudy weather shows less variation. To study the variation in fluctuation temperature (*T_f_*) on a daily basis, the *T_f_* values can be classified according to sunny and cloudy days as well as cold and hot seasons. Since the fluctuations of cloudy days are more moderate, this study focuses on the analysis of sunny days. Thus, the *T_f_* results are divided into two categories: typical sunny day data for the hot season and typical sunny day data for the cold season.

The measured *T_f_* values of several typical sunny days in the hot season are selected as examples for demonstration. The *T_f_* data of typical sunny days are listed separately by the day cycle, as depicted in Figure 10a. As seen, the trend of *T_f_* values on different days essentially stays the same and the only difference is the changing amplitude. A top view of Figure 10a is shown in Figure 10b. During the sunny days of a hot season, it is visible that *T_f_* has the highest amplitude during the time of 13:00–15:00 and the lowest amplitude during the time of 5:30–7:00.

In the illustration, the same sample of measured *T_f_* from several typical sunny days during the cold season is given. The *T_f_* data of typical sunny days are listed separately by the day cycle in Figure 11a. As seen, the trend of *T_f_* on different days essentially stays the same, and the only difference is the changing amplitude. From Figure 11b, it is visible that *T_f_* has the highest amplitude during the time of 13:00–15:00 and the lowest amplitude during the time of 6:00–7:30. Additionally, the occurrence time of extreme *T_f_* values (i.e., maximum and minimum) was counted separately for all sunny days during the hot and cold seasons, as summarized in Table 1.

Table 1 shows that the highest *T_f_* occurs between 13:00 and 15:00 hours, covering 70.49% of all hot season data, and that the lowest *T_f_* occurs between 5:30 and 7:00 hours, covering 65.57% of all hot season data. Similar conclusions can be drawn for the cold season. Therefore, based on the statistical results, the minimum fluctuating temperature appears at a more scattered time, while the maximum fluctuating temperature appears at a more concentrated time. In general, the lowest *T_f_* occurs at the time of 5:30–7:30; the highest *T_f_* occurs at the time of 13:00–15:00.

#### 3.4.2. Theoretical Method of Extreme Value Analysis

From the statistical point of view, the actual maximum *T_f_* of the cable may exceed the test values. Thus, in bridge designs, the extreme *T_f_* values are generally adopted as the representative values, which can be determined by the extreme value analysis (EVA).

The EVA is a method that has been used in many fields, such as engineering and economics, and focuses on the behavior of extreme observations rather than the entire sample population [35]. The generalized extreme value (GEV) distributions and maximum entropy (MaxEnt) distributions [36] have been widely applied in the EVA. Compared with the GEV, the MaxEnt model is more stable and significantly more robust in the variation in sample sizes, indicating that the MaxEnt model reduces the outcome uncertainty and avoids the high risk of bias. This robustness of the MaxEnt for the sample sizes has also been demonstrated by Dai et al. [37]. Therefore, the MaxEnt model is adopted in this paper to determine the extreme temperatures of cables.

According to Dai et al. [37], the MaxEnt model is expressed as
(6)$$\max H=-\underset{D}{\int }\ln(f(x))f(x)dx$$
where ln represents the natural logarithm and *f*(*x*) is a probability distribution function that satisfies the following constraint conditions:(7)$$\underset{D}{\int }f(x)dx=1$$
(8)$$\underset{D}{\int }x^{i}f(x)dx=m_{i},\;i=1,2,\cdots ,r$$
where *D* is the integration interval, *r* is the order of the moments used, and *m_i_* is the *i*-th order origin moment of the sample. Then, by introducing the Lagrangian function and corresponding multipliers *λ_i_*, the expression of the MaxEnt probability density function (PDF) is obtained as
(9)$$f(x)=\exp\left(\lambda_{0}+\sum_{i=1}^{r}\lambda_{i}x^{i}\right),\;i=1,2,\cdots ,r$$
whose cumulative distribution function (CDF) is
(10)$$F(x)=\underset{D}{\int }\exp\left(\lambda_{0}+\sum_{i=1}^{r}\lambda_{i}x^{i}\right)dx,\;i=1,2,\cdots ,r$$

The PDF *f*(*x*) is obtained by solving the Lagrangian multipliers from the nonlinear system with (*r* + 1) equations. The improved Newton iterative method [38], which is an efficient and accurate estimation technique, is adopted in this study to solve the PDF *f*(*x*) of extreme temperatures.

#### 3.4.3. Extreme Value Analysis of Daily Cycle Variation

According to the Chinese specification [27], for bridge thermal action with an exceedance probability of 0.01, the probability, *p*, of *T_f_* exceeding the extreme value, can be set as 0.01 (equivalent to a mean return period of 100 days). To obtain a representative daily fluctuating temperature, *T_f_*, at each moment, the collected *T_f_* at a certain moment of the day in the hot season (or cold season) can be taken to constitute the sample set. After fitting the data with the maximum entropy extreme value model, the representative *T_f_* values for each time are determined. After that, a fluctuation temperature envelope curve with an exceedance probability of 0.01 is generated using the representative *T_f_* values at each moment.

Taking the hot season as an example, as seen in Figure 10b, the maximum variation amplitude of *T_f_* occurs around the hour of 13:00. Thus, the sample group can be formed by taking the *T_f_* at 13:00 hour on sunny days during the hot season. The ManEnt extreme value model is used to statistically analyze this sample group and the histogram of the 13:00-hour sample group, and the fitted probability density function (PDF) and cumulative probability function (CDF) are obtained and plotted, as shown in Figure 12a. The density distribution plot is consistent with the histogram of the data. Furthermore, the probability-probability (P-P) plot technique, which plots the empirical CDF against the theoretical CDF, is employed to better illustrate the fitness of the MaxEnt extreme value model, as shown in Figure 12b.

As shown in Figure 12b, the points in the P-P plot approximately lie on the 45° comparison line from (0,0) to (1,1). As seen, the MaxEnt model fits the data well. Then, to obtain representative *T_f_* values at the hour of 13:00, the CDF of the 13:00-hour sample group is obtained after the fitting of the MaxEnt model, as shown in Figure 13. The probability that *T_f_* exceeds the extreme value by +0.01 point is 0.01 and the probability that it is lower than the extreme value by the −0.01 point is also 0.01. These two points are taken as the representative values of *T_f_* at 13:00 hour. With the data collected every half hour, a day can be split into 48 sample groups for analysis. Aside from the 13:00-hour sample group, the remaining 47 sample groups are subjected to the same extreme value analysis procedure, and the MaxEnt model produces a comparable level of fit goodness. As shown in Figure 14, all obtained representative *T_f_* values at each moment are plotted onto a fluctuation temperature envelope curve with an exceedance probability of 0.01. The representative *T_f_* values at each moment are connected with a line to form the upper and lower boundaries of the daily cycle variation, whereas the scattered points represent the sample of collected *T_f_* values for all sunny days during the hot season. The sample points fall within the upper and lower bounds. Therefore, it is reasonable to use the fluctuation temperature envelope curve with an exceedance probability of 0.01 to represent the daily periodic variation in cable temperatures. During the hot season, *T_f_* is the lowest around 6:00 hour and highest around 13:00 hour, and the amplitude of the upper and lower bounds is 11.8 °C. Under the condition of exceeding probability = 0.01, the range of *T_f_* variation is −4.56~7.28 °C.

With an exceedance probability of 0.01, the same fluctuation temperature envelope curve can be concluded from the *T_f_* data collected during the cold season. The sample group can be formed in the same way as the previous calculation by taking the *T_f_* at 13:00 hour on sunny days during the cold season. The MaxEnt extreme value model is used to statistically analyze this sample group; the histogram of the 13:00 sample group and the fitted PDF and CDF are obtained and plotted in Figure 15a. The density distribution plot is consistent with the histogram of the data. Then, the information represented by the P-P plot indicates a good fit, as shown in Figure 15b. The points in the P-P plot approximately lie on the 45° comparison line from (0,0) to (1,1). As noted, the MaxEnt model fits satisfactorily with the 13:00-hour sample group. Then, the CDF of the 13:00-hour sample group is obtained, as shown in Figure 16. The probability that *T_f_* exceeds the extreme value by +0.01 point is 0.01, and the probability that it is lower than the extreme value by −0.01 point is also 0.01. These two points are taken as the representative values of *T_f_* at 13:00 hour. The remaining 47 sample groups, aside from the 13:00-hour group, are subjected to the same extreme value analysis procedure, and the MaxEnt model produces a comparable level of fit goodness. The representative *T_f_* values at each moment are connected with a line to form the upper and lower boundaries of the daily cycle variation, whereas the scattered points represent the sample of collected *T_f_* values for all sunny days during the cold season, as shown in Figure 17. During the cold season, *T_f_* is lowest around 6:30 hour and highest around 15:00 hour, and the amplitude of the upper and lower bounds is 15.3 °C. Under the condition of exceeding probability = 0.01, the range of *T_f_* variation is −6.22~9.15 °C.

Therefore, based on the above analysis, the daily cycle of the temperature variation in the cable can be represented by the fluctuation temperature envelope curve with an exceedance probability of 0.01, with *T_f_* varying from −6.22 °C to 9.15 °C in the cold season and from −4.56 °C to 7.28 °C in the hot season. As observed, the variation in cable fluctuating temperature is more noticeable throughout the year, and the variation amplitude of *T_f_* varies more in the cold season than in the hot season. The variation in *T_f_* reflects the expansion and contraction of the cable in a single day. Therefore, to accurately determine the temperature deformation of the cable, it is important to consider both the daily temperature fluctuations (*T_f_*) and the cable’s annual cycle of uniform temperatures (*T_u_*).

## 4. Relationship between the Cable Temperature and Environmental Variables

After analyzing the daily cycle variation pattern to obtain the fluctuation temperature (*T_f_*) envelope curve with an exceedance probability of 0.01, the *T_u_* representative value of the cable annual cycle variation pattern is further investigated. Thus, in bridge designs, the extreme *T_u_* values are generally adopted as the representative values, which can be determined by statistical analysis. According to the 100-year design reference period for bridges in the Chinese specification [27] and the 50- and 100-year return periods of thermal action assumed in this study, the probability, *p*, for *T_u_* exceeding the extreme value can be calculated as
(11)$$p=\frac{1}{100\times n}$$
where *n* is the number of data in a full year. If the annual extreme value is calculated, *n* equals 1, giving *p* = 0.02 for the 50-year return period and *p* = 0.01 for the 100-year return period.

Annual extremes are the ideal samples in extreme value analysis to determine representative values. However, only one annual extreme temperature effect was available in this measurement experiment, making it difficult to meet the sample size requirements for extreme value analysis. Therefore, a machine learning approach is further employed to establish the relationship between *T_u_* and environmental variables for simulating the annual extreme values of the temperature action to supplement the data. Some researchers [39] have tried to simulate the cable temperatures based on the air temperature and solar radiation using heat-transfer analysis, in which a finite element model conforming to complex environmental boundary conditions is established. However, the input parameters for such models are difficult to determine, so they are usually estimated. Furthermore, the FE analysis requires costly computations to obtain a structural temperature model. Therefore, it is necessary to study the relationship between the cable temperature and the environmental variables and explore a method to obtain the cable temperature directly from the actual measured data relating to the environmental variables. This study uses machine learning techniques to establish the relationship between multiple factors and uniform temperature, which are well suited for extracting knowledge, information, and correlations from data.

### 4.1. Regression via the Gradient Boosted Regression Trees (GBRT) Algorithm

#### 4.1.1. Description of the GBRT Algorithm

The GBRT algorithm is an ensemble learning algorithm that uses multiple regression trees as the base predictors [40,41]. The GBRT works by sequentially adding the predictors to an ensemble, with each correcting its predecessor. Therefore, every regression tree is not self-contained, since this method tries to fit the new regression tree to the residual errors made by the previous regression tree. Figure 18 depicts the GBRT algorithm. Noted that the residual of former regression trees is used as the training data for the following regression tree. The added regression tree is then used to minimize the residuals of the previous trees, resulting in decreasing losses in each iteration with the negative gradient direction developed. Finally, the prediction result is determined based on the sum of all regression tree outcomes.

Let ${\left\{x_{i},y_{i}\right\}}_{i=1}^{m}$ denote the sample data, where *x_i_* = (*x_*1*i_*, *x_*2*i_*,…, *x_ri_*) represents the indicators, *r* is the number of the predicted variables, and *y_i_* is the predicted label. The steps of the GBRT algorithm are described as follows.

Step 1: The initial constant value of *γ* is obtained by
(12)$$F_{0}(x)=\arg\underset{\gamma }{\min}\sum_{i=1}^{m}L(y_{i},\gamma )$$
where *L*(*y_i,γ_*) is the loss function.

Step 2: In the gradient boosting algorithm, the gradient descent is used to reduce errors. The residual along the gradient direction is expressed as
(13)$${\overset{⏜}{y}}_{i}=-{\left[\frac{\partial L\left(y_{i},F(x_{i})\right)}{\partial F(x_{i})}\right]}_{f(x)=f_{n-1}(x)}$$
where *n* is the number of iterations (*n* = 1, 2, …, *N*).

Step 3: The initial constant model *T_i_*(*x_i_*;*α_n_*) is obtained by fitting the sample data, and the parameter *α_n_* is calculated based on the least square method as
(14)$$a_{n}=\arg\underset{\alpha ,\gamma }{\min}{\sum_{i=1}^{m}\left({\overset{⏜}{y}}_{i}-\gamma T_{n}(x_{i};\alpha )\right)}^{2}$$

Step 4: By minimizing the loss function, the current model weight is expressed as
(15)$$\gamma_{n}=\arg\underset{\gamma }{\min}\sum_{i=1}^{m}L(y_{i},F_{n-1}(x)+\gamma T_{n}(x_{i};\alpha_{n}))$$

Step 5: The model is updated as
(16)$$F_{n}(x)=F_{n-1}(x)+\gamma_{n}T_{n}(x_{i};\alpha_{n})$$

The above steps are performed until the specified number of iterations or the convergence condition is met.

#### 4.1.2. Construction of the Prediction Model

The construction process of the prediction model is shown in Figure 19. The basic concept is to predict the cable uniform temperature, *T_u_*, based on the air temperature (*T_air_*), solar radiation intensity (SRI), wind speed (WS), and wind direction (WD). To evaluate the accuracy of a machine learning model, the dataset is usually divided into two subsets: training and test. First, approximately 80% and 20% of the original measured data are selected as training and test datasets, respectively. Second, a five-fold cross-validation (CV) approach [42] is used to fine-tune the model hyperparameters, and the GBRT model is fitted based on the training set using the optimal hyperparameters configuration. Then, the GBRT model is capable of predicting the cable uniform temperature, *T_u,_* based on multi-environment factors. Third, the test set is used for the prediction. Therefore, the *T_air_*, SRI, WS, and WD are input into the GBRT model. The result is the predicted cable uniform temperature, *T_u,predicted_*. Last, the performance of the model with unknown data is evaluated by comparing the measured and the predicted data. If the prediction performance of this model is acceptable, then it can be adopted for deployment.

### 4.2. Training and Testing of T_u_ Prediction Model

The *T_u_* prediction model is constructed in this study using the GBRT algorithm. The model inputs include the four environmental variables: *T_air_*, SRI, WS, and WD. The entire calculation is performed in Python 3.7 based on the scikit-learn [43]. Hyperparameters are vital for the GBRT, including n_estimators (i.e., number of trees), learning_rate (i.e., shrinkage coefficient of each tree = 0~1), and max_depth (i.e., tree complexity).

The measured data for the cable used in the GBRT model are the *T_u_* obtained from the previous sectional analysis. The data for the environmental variables are obtained from the China Meteorological Center over a time span of 34 years. Before training the model, the data needs to be processed in two main steps: (1) outliers and missing values are cleaned from measured and meteorological data, and (2) the meteorological data for the time period from September 2020 to September 2021 are centrally extracted from the downloaded meteorological data, and the measured temperature data of the cable correspond to the respective environmental variables according to the time and form a sample set.

The following step is to perform featured engineering on such a sample set. Featured engineering is the process of extracting data features from raw data that are relevant to the problem and transforming them into a data set of input features that can be recognized by the algorithm. The purpose of featured engineering is to help the algorithm learn better from the data in order to achieve the desired prediction results. The GBRT will only be capable of learning if the training data contains enough relevant features and few irrelevant ones. The success of a machine learning project depends on a good set of features for training. Based on the input features and meteorological data characteristics of the GBRT model, the featured engineering factors listed in Table 2 are considered in this study.

### 4.3. Hyperparameter Optimization

The data set is divided into two subsets: training (80%) and test (20%) sets. The training set is used to estimate model parameters and the test set is used to check the generalization ability of the model. To evaluate the performance of the GBRT model, a devised metric is needed to quantify the advantages and disadvantages of the model. The coefficient of determination (*R*^2^) and the mean absolute error (MAE) are chosen as the model performance indicators. The smaller the MAE, the better the model; the larger *R*^2^ values, the better the model. The following equations are used to determine the parameter.

Mean absolute error:(17)$$\mathrm{MAE}=\frac{1}{n}\sum_{i=1}^{n}\left|y_{i,real}-y_{i,predicted}\right|$$

Coefficient of determination:(18)$$R^{2}=1-\frac{\sum_{i=1}^{n}{\left(y_{i,real}-y_{i,predicted}\right)}^{2}}{\sum_{i=1}^{n}{\left(y_{i,real}-\bar{y}\right)}^{2}}$$
where *y_i,predicted_* is the predicted value of the *i*-th sample, and *y_i,real_* is the corresponding true (measured) value over *n* samples. $\bar{y}=\frac{1}{n}\sum_{i=1}^{n}y_{i,real}$.

To achieve optimum parameter values and a robust outcome, the grid search method is applied in the GBRT algorithm. The grid search means that given a search interval of model parameters, all possible combinations within each parameter interval are generated, and then the model is trained on the data. Based on the performance of the model on the training set, the optimal combination of hyperparameters is selected. The critical hyperparameters in the GBRT algorithm are identified, as listed in Table 3, where the specific meanings, the search ranges, and the optimal values are indicated.

### 4.4. Testing of the GBRT

The test set is used for prediction. The input multiple environmental variables in the test set are shown in Figure 20. Therefore, the air temperature, SRI, WS, and WD are input into the GBRT model. The cable uniform temperature, *T_u_*, is predicted, as shown in Figure 21. The model is applied during the entire measurement period (1 year). There is no noticeable difference between the training set (80%) and the test set (20%), indicating that the model does not overfit (Table 4). All metrics are close to each other for the different datasets.

The graphs of the measured and the predicted cable uniform temperatures agree very well (Figure 21a). The histogram for the differences between the measured and the predicted cable uniform temperatures is plotted in Figure 21b along with the distribution of residuals. As noted, the residuals are basically concentrated in the interval of −1 °C to 1 °C. In addition, *R*^2^ > 0.98 and the MAE < 0.4 for the test set, as indicated in Figure 21b. The results manifest that the GBRT method is suitable for the present problem.

To compare the performance of the GBRT model and the conventional univariate linear regression (ULR) model, the samples in the training set are used to fit the ULR models of *T_u_*. Air temperature is used as the predictor of the ULR models, as shown in Equation (14). The measured *T_u_* values are compared with the predicted values using the GBRT and ULR models on the test set, as shown in Figure 22. The test *R*^2^ of the ULR model is 0.94, while it is above 0.98 for the GBRT model with multiple environmental factors as the predictors. The better prediction accuracy from the GBRT model is particularly evident for high *T_u_* values (Figure 22).
(19)$$T_{u}=1.059T_{air}-1.638\;R^{2}=0.94$$

### 4.5. Representative Values of Temperature Actions of Cables

To obtain the representative value of the temperature actions of cables for design, a long-term observation of the structure’s temperatures is required, from which a representative temperature value can be obtained by the EVA method. In practical situations, it is often difficult to obtain high-quality long-term observations of structural temperatures. Nevertheless, the collection of long-term data on meteorological variables such as air temperature and solar radiation intensity is possible. Therefore, by adding the long-term historical meteorological data to the previously trained GBRT model, the multi-year cable temperature data can be obtained. Then, the MaxEnt model is used to determine the representative values of the temperature actions of bridge cables.

The meteorological data collected from the Meteorological Center spanning 34 years, from 1987 to 2020, in the Quanzhou area (bridge site), a total of about 290,000 meteorological data are obtained, which contain the same environmental variables as described above. Using the obtained meteorological data of 34 years, the featuring work is conducted as listed in Table 3, and then the trained GBRT model is used to calculate the uniform temperature, *T_u_*, of the cable from the 34-year data.

A diagram of the block maxima (BM) is given in Figure 23, where the blocks are chosen for a period of one year. According to the MaxEnt model, even if the distribution of the data is unknown, the distribution of annual extreme values can be obtained by fitting with the MaxEnt model. Therefore, the extreme values are selected from the 34 sets of data, as shown in Figure 23, followed by repeated sampling 34 times to obtain 34 annual extreme data points (Table 5). Finally, with the obtained data, the MaxEnt distribution is established, and the representative statistical extreme value of the temperature action is determined for a given return period.

A design life of 100 years for bridges is stipulated in the Chinese Code Specification [27], with a 50-year return period assumed in Eurocode 1 [28] and used in the EVA. The 50- and 100-year return periods of thermal action are assumed in this study and the exceeding probability values, *p*, are calculated to be 0.02 and 0.01.

Figure 24 gives the result of fitted MaxEnt distribution of annual maximum uniform temperature (*T_u,max_*) and the statistical extreme values for 50-year/100-year return periods. The frequency histograms and fitted curves (PDF and CDF) are plotted in Figure 24a. The MaxEnt model is verified to have the fine accuracy of fitting by P-P tests. As seen in Figure 24b, the points approximately lie on the 45°comparison line from (0,0) to (1,1), which indicates a good fit.

Similar to the previous example, the MaxEnt model is also applied to construct the distribution of the annual minimum uniform temperatures (*T_u,min_*). The fitted PDF is consistent with the histogram of the data, as shown in Figure 25a. Furthermore, the set of plotted points is nearly linear, as shown in Figure 25b, which validates the fitted model.

For design purposes, a representative value for each temperature action is desirable. The detailed procedures for that are as follow: (1) the maximum, *T_u,max_*, and the minimum, *T_u,min_*, are applied to the statistical extreme values of $T_{u,max}^{50/100\;yr}$ and $T_{u,min}^{50/100\;yr}$; (2) all statistical extreme temperature actions are constructed by the MaxEnt model, and the prediction of *T_u_* can be done by adopting different return periods; and (3) taking the return periods of 50 and 100 years, the predicted extreme values ($T_{u,max}^{50/100\;yr}$ and $T_{u,min}^{50/100\;yr}$) are then taken as the representative values.

Figure 26 gives the cumulative probability function of annual maximum and minimum uniform temperatures of the cable, respectively. All the representative values of uniform temperature actions of cables are listed in Table 6. For this stayed-cable, the maximum uniform temperatures are 38.82 °C and 39.51 °C and the minimum uniform temperatures are 1.12 °C and 0.27 °C for the 50-year and 100-year return periods, respectively.

## 5. Conclusions

This study investigates the issue of cable temperatures on a long-span crossing-sea high-speed rail bridge, where the methods of segment test, machine learning, and extreme value analysis (EVA) are adopted. First, the temperature distribution pattern and time-history curve of the cable are investigated, the annual cycle variation in the uniform temperature (*T_u_*) of the cable is established, and the fluctuation temperature (*T_f_*) envelope curve of the daily cycle variation in the cable is derived. Then, using the GBRT method, the relationship between the cable temperature and multiple environmental variables is explored, and the design representative values of the cable uniform temperature are obtained by a statistical analysis. Based on this study, the following conclusions can be drawn:No temperature gradient distribution is noted inside the cable, suggesting a uniform temperature distribution on the cross-section. The temperature on the cable cross-section can be decomposed into a daily uniform temperature (*T_u_*) and a daily fluctuating temperature (*T_f_*) to portray the annual cycle and daily cycle variation patterns of cable temperatures, respectively.For the annual cycle variation pattern, the time-history curve of *T_u,d_* (daily average uniform temperature) can be fitted to the Fourier formula. Due to the effect of *T_u,d_*, the cable contracts and expands seasonally. The daily cycle of temperature variations can be represented by the fluctuation temperature envelope curve with an exceedance probability of 0.01, and the variation in the cable’s fluctuating temperatures is more noticeable throughout the year. Therefore, to accurately determine the temperature deformation of the cable, it is important to consider both the daily temperature fluctuations and the annual cycle of uniform temperatures.This study establishes the use of the GBRT model to predict *T_u_* in the cable considering four environmental variables (air temperature, solar radiation intensity, wind speed, and wind direction) at the bridge site. As for *T_u_* prediction, the prediction accuracy of the GBRT model is significantly higher than that of the univariate linear regression (ULR) model that considers a single environmental factor only.A complete EVA method based on the machine learning simulation and one-year measured data is proposed for determining the representative values of temperature effects. This method can compensate for the deficiency of measured data and determine the representative values of bridges’ temperature effects quickly and efficiently. For the measured stayed-cable segment, the representative maximum uniform temperatures are 38.82 °C and 39.51 °C and the representative values of the minimum uniform temperature are 1.12 °C and 0.27 °C for the 50- and 100-year return periods, respectively.

The results concluded from this study shall provide a good basis for the operation and maintenance of in-service long-span cable-stayed bridges.

## Figures and Tables

**Figure 1 sensors-23-05675-f001:**
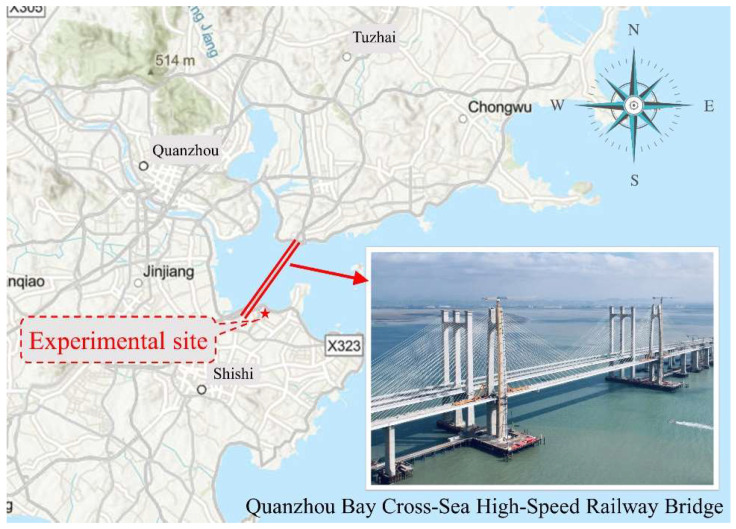
Location of the experimental site.

**Figure 2 sensors-23-05675-f002:**
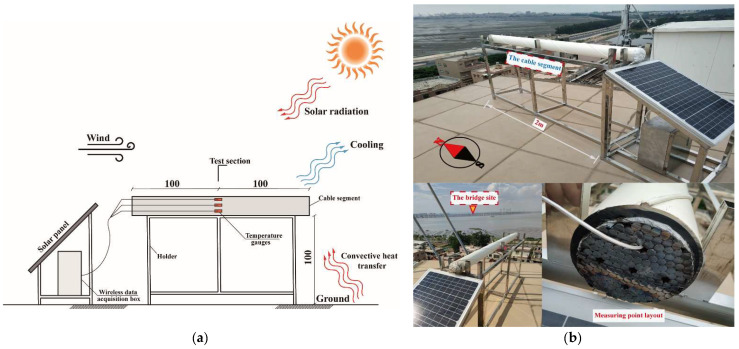
Diagram of the experimental cable segment (mm). (**a**) Schematic diagram of the experimental cable segment. (**b**) Photo of the experimental cable segment.

**Figure 3 sensors-23-05675-f003:**
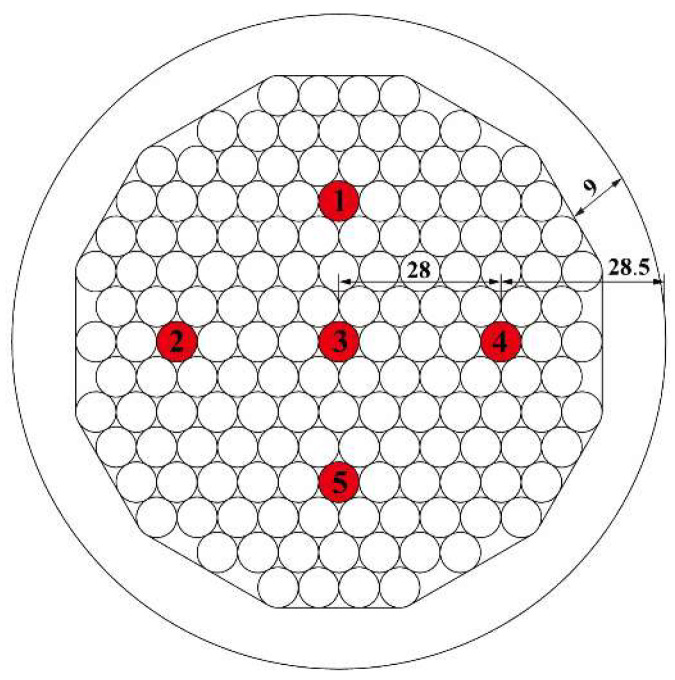
The geometry of the cable cross-section and the locations and numbing of temperature sensors (Number the measurement points and fill them in red; Unit: mm).

**Figure 4 sensors-23-05675-f004:**
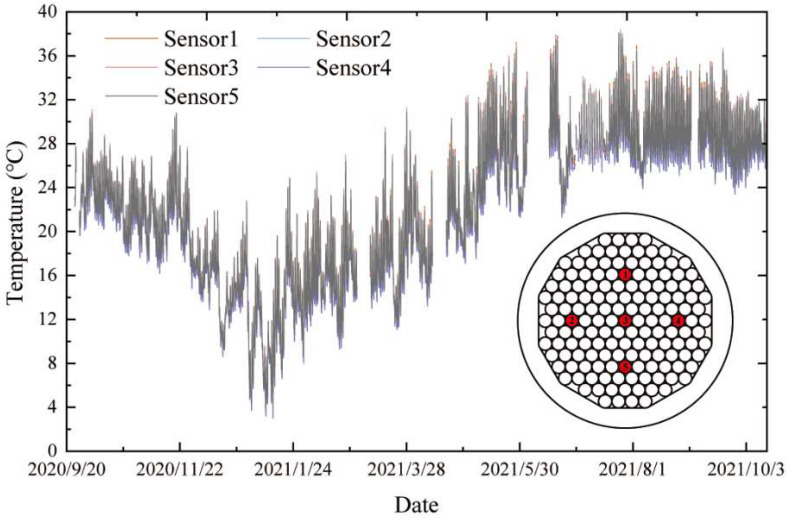
The measured data of cable segment test points.

**Figure 5 sensors-23-05675-f005:**
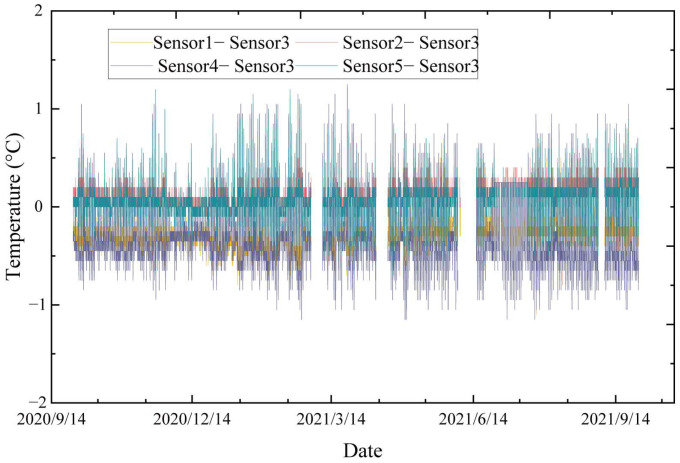
Temperature differences among the measuring points.

**Figure 6 sensors-23-05675-f006:**
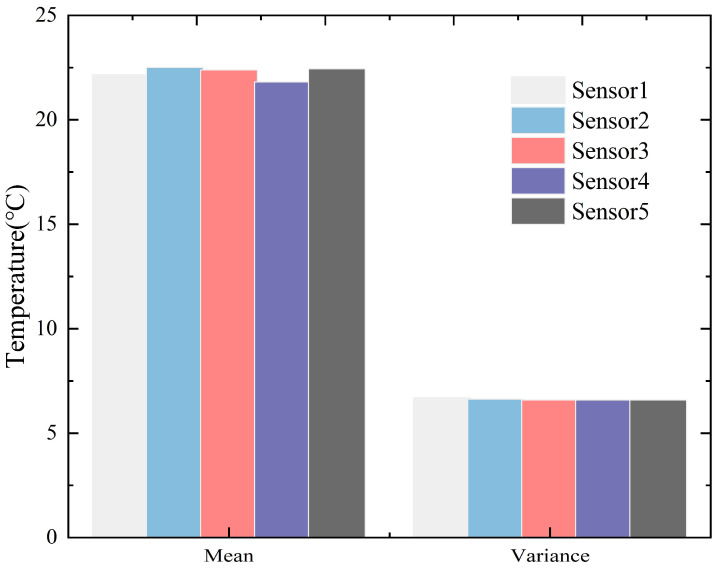
Mean and variance of the temperature data at each measuring point.

**Figure 7 sensors-23-05675-f007:**
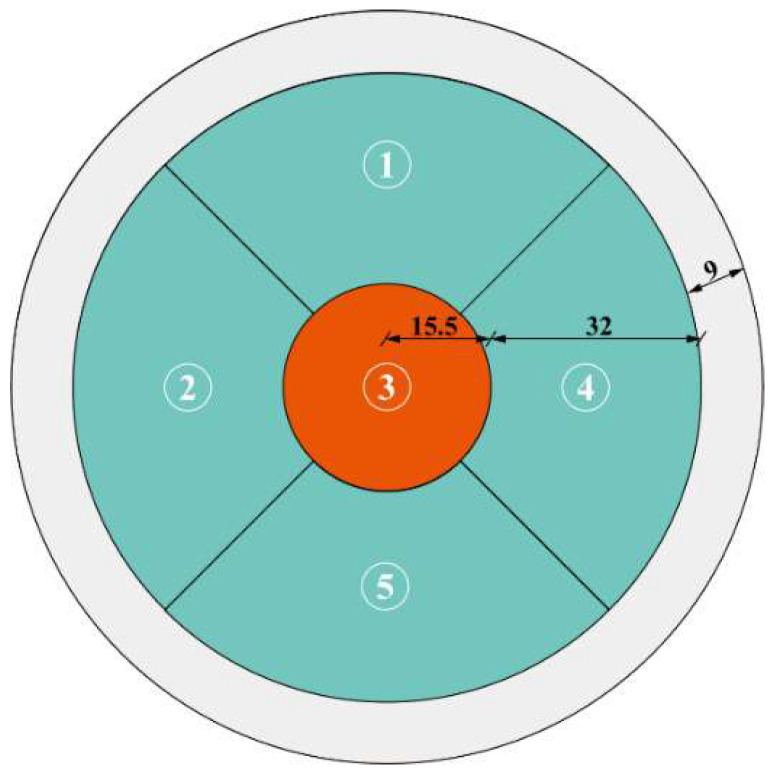
Typical cross-section of a partition (mm).

**Figure 8 sensors-23-05675-f008:**
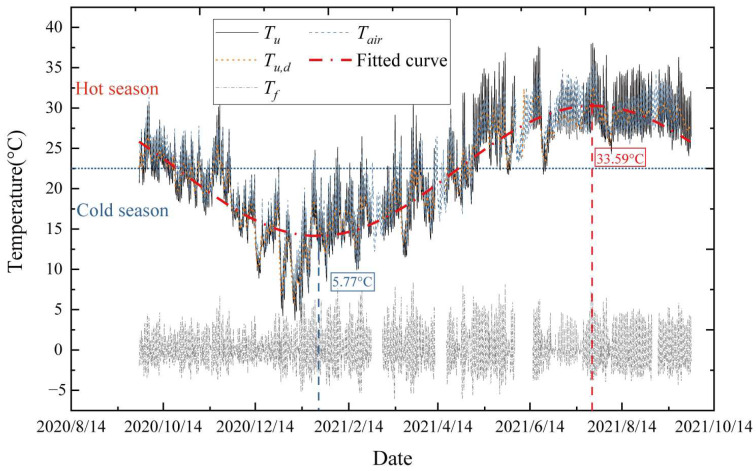
Time histories of the uniform temperature and fluctuating temperature of the cable.

**Figure 9 sensors-23-05675-f009:**
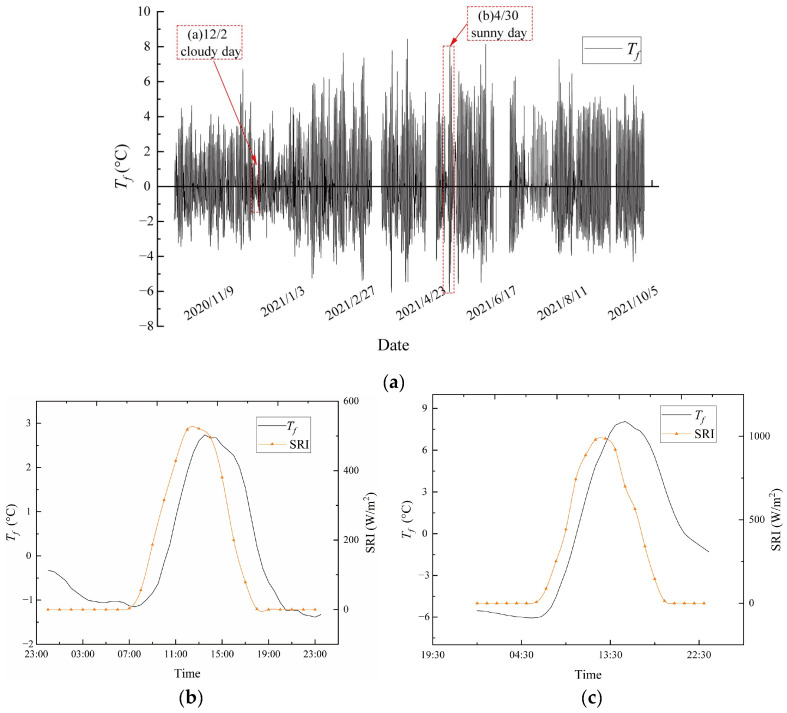
Time history of the fluctuating temperatures, *T_f_*, of the cable. (**a**) Time history of the daily fluctuation *T_f_*; (**b**) *T_f_* and SRI on 2 December 2020 (a cloudy day); (**c**) *T_f_* and SRI on 30 April 2021 (a sunny day).

**Figure 10 sensors-23-05675-f010:**
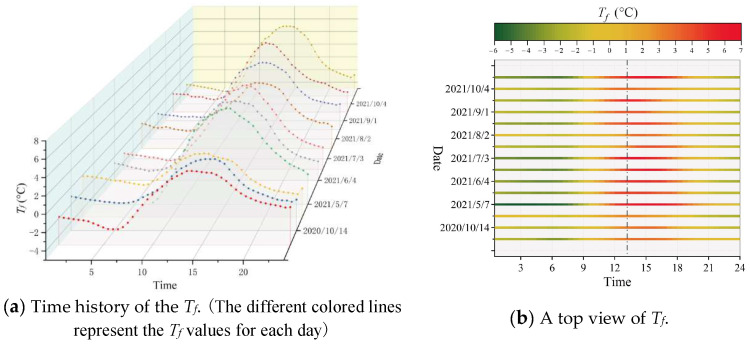
Time history of the fluctuating temperature, *T_f_*, of typical sunny days in hot season.

**Figure 11 sensors-23-05675-f011:**
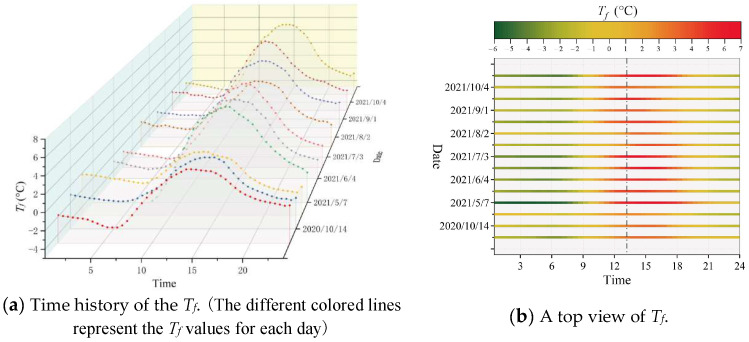
Time history of the fluctuating temperature, *T_f_*, of typical sunny days in cold season.

**Figure 12 sensors-23-05675-f012:**
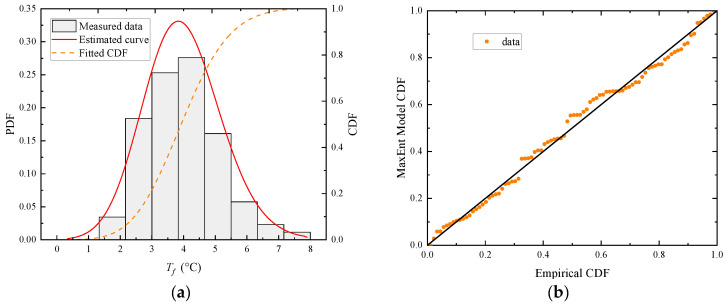
Diagnostic plots of the MaxEnt model fitted to the 13:00-hour *T_f_* sample group on sunny days during the hot season. (**a**) The 13:00-hour *T_f_* sample group density plot; (**b**) the 13:00-hour *T_f_* sample group probability plot.

**Figure 13 sensors-23-05675-f013:**
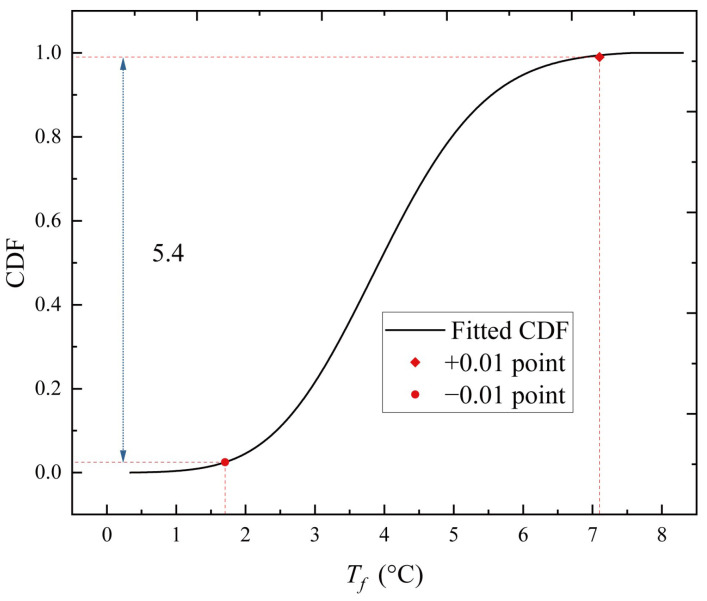
CDF of 13:00-hour *T_f_* sample group on sunny days during the hot season.

**Figure 14 sensors-23-05675-f014:**
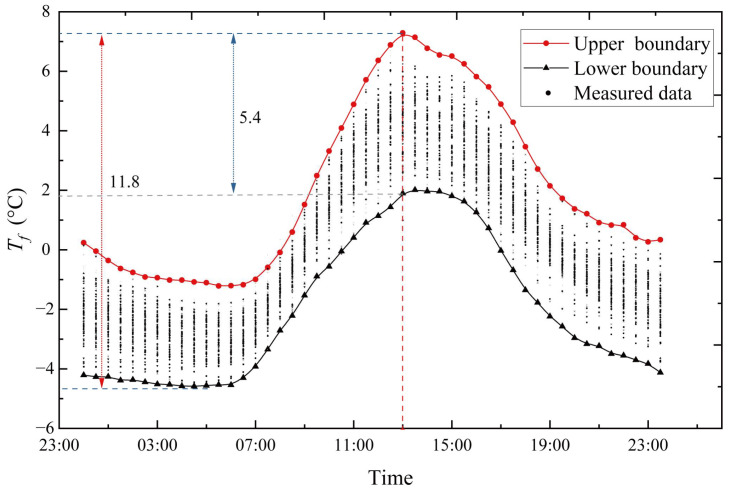
Fluctuation temperature envelope curve with an exceedance probability of 0.01 on sunny days during the hot season.

**Figure 15 sensors-23-05675-f015:**
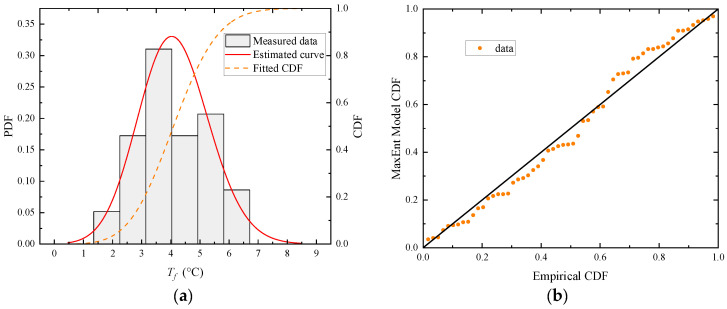
Diagnostic plots of the MaxEnt model fitted to the 13:00-hour *T_f_* sample group on sunny days during the cold season. (**a**) The 13:00-hour *T_f_* sample group density plot; (**b**) the 13:00-hour *T_f_* sample group probability plot.

**Figure 16 sensors-23-05675-f016:**
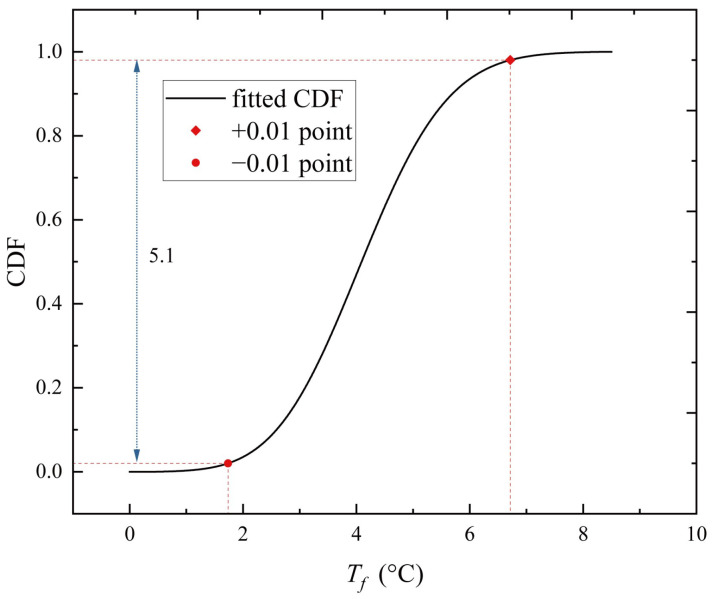
CDF of 13:00-hour *T_f_* sample group on sunny days during the cold season.

**Figure 17 sensors-23-05675-f017:**
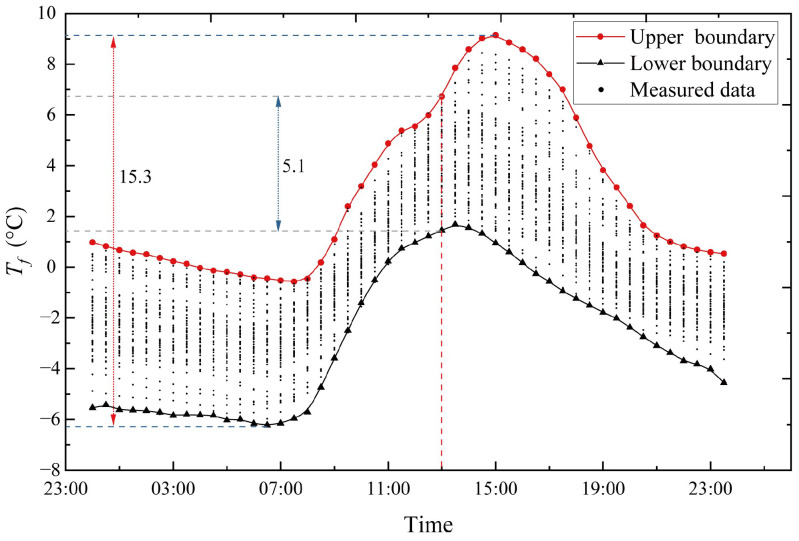
Fluctuation temperature envelope curve with an exceedance probability of 0.01 on sunny days during the cold season.

**Figure 18 sensors-23-05675-f018:**
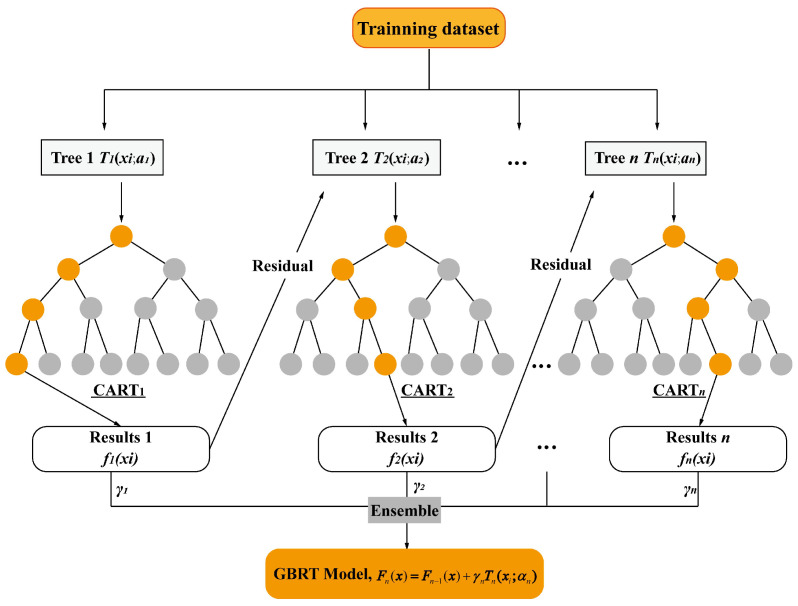
Schematic diagram of the GBRT algorithm.

**Figure 19 sensors-23-05675-f019:**
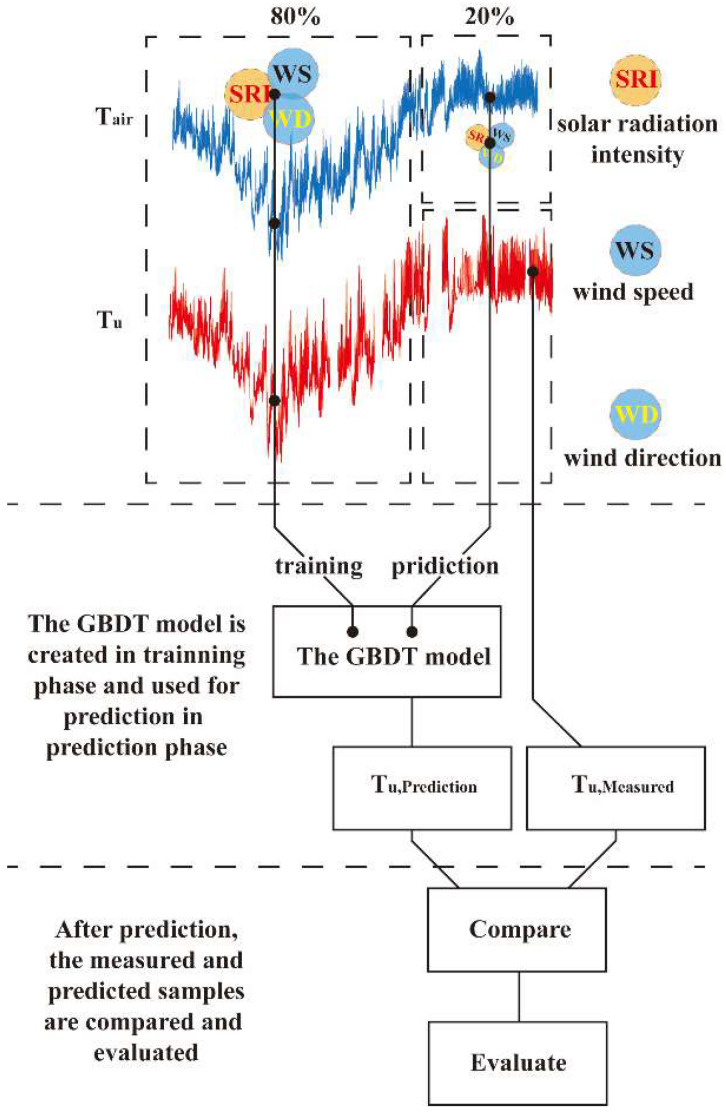
Flowchart of the GBRT algorithm.

**Figure 20 sensors-23-05675-f020:**
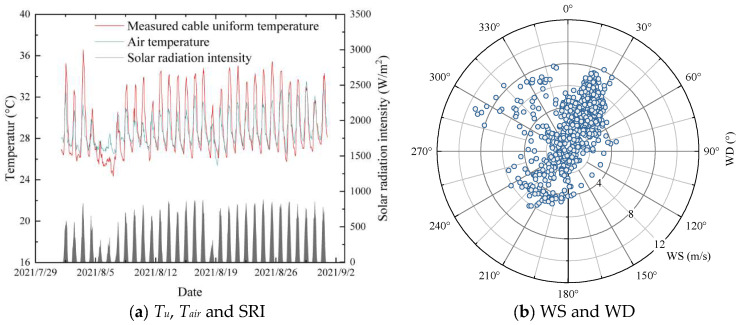
Input of the GBRT algorithm.

**Figure 21 sensors-23-05675-f021:**
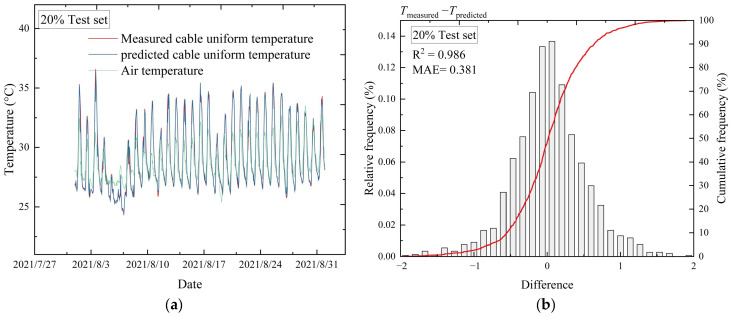
Comparison between measured and predicted cable uniform temperatures in the test set. (**a**) Comparison between the predicted results and the measured cable uniform temperatures. (**b**) Histogram for the differences between measured and predicted cable uniform temperatures.

**Figure 22 sensors-23-05675-f022:**
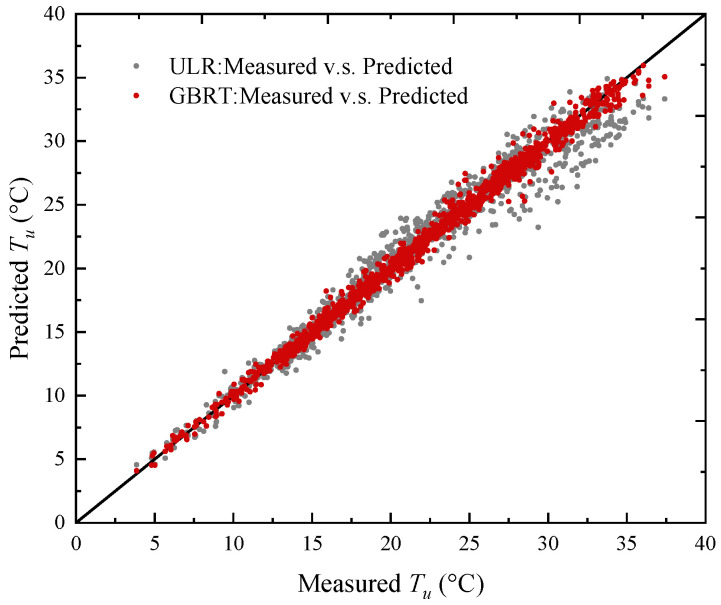
Comparison between the measured and the predicted values of *T_u_* on the test set. (ULR = univariate linear regression; GBRT = gradient boosting regression trees).

**Figure 23 sensors-23-05675-f023:**
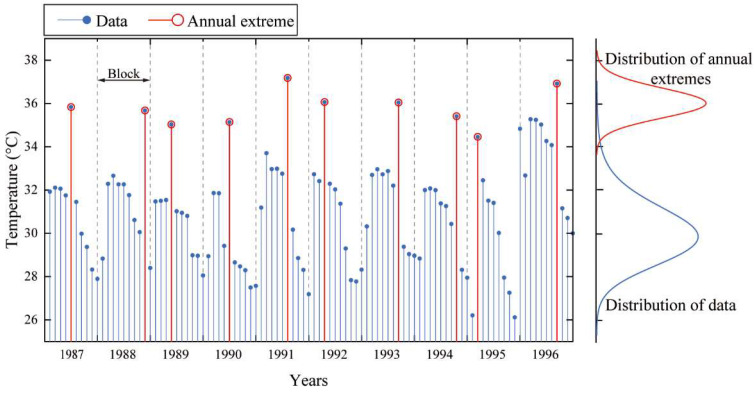
Diagram of annual extremes.

**Figure 24 sensors-23-05675-f024:**
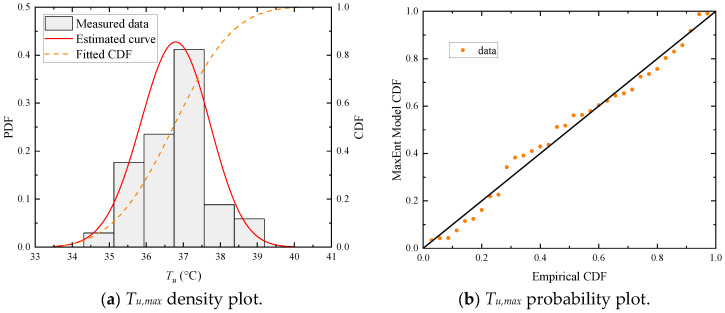
Fitted distribution of annual maximum uniform temperatures (*T_u,max_*).

**Figure 25 sensors-23-05675-f025:**
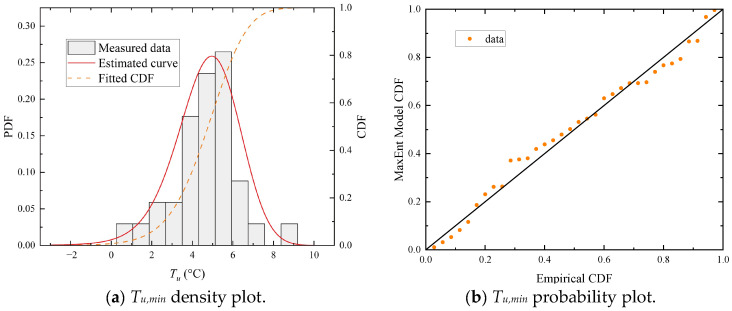
Fitted distribution of annual minimum uniform temperatures (*T_u,min_*).

**Figure 26 sensors-23-05675-f026:**
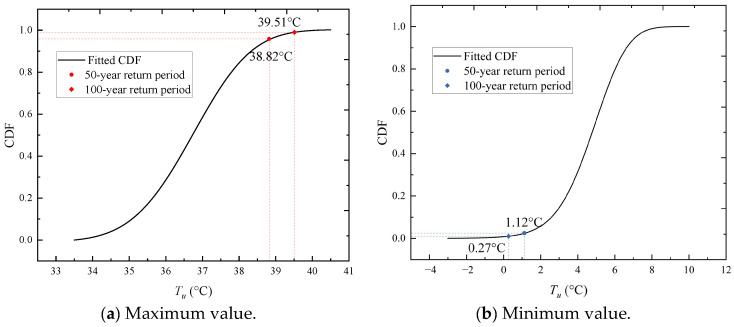
CDF of annual uniform temperature extremes of the cable.

**Table 1 sensors-23-05675-t001:** Summary of fluctuating temperature, *T_f_*, of typical sunny days.

Season	Minimum Fluctuating Temperature, *T_f_*		Maximum Fluctuating Temperature, *T_f_*	
	Main Time of Occurrence	Percentage of All Data	Main Time of Occurrence	Percentage of All Data
Hot season	5:30–7:00	65.57%	13:00–15:00	70.49%
Cold season	6:00–7:30	51.87%	13:00–15:00	73.76%

**Table 2 sensors-23-05675-t002:** Featured engineering of the GBRT.

Number	Featured Engineering	Units
1~48	Air temperature for the past 48 h	°C
49~96	Solar radiation intensity for the past 48 h	W/m^2^
97~144	Wind speed for the past 48 h	m/s
145~192	Wind direction for the past 48 h	°
193	Date	-
194	Time	-

**Table 3 sensors-23-05675-t003:** Hyperparameters optimization results of the GBRT.

Hyperparameters	Meanings	Search Ranges	Optimal Values
n_estimators	Number of trees	(150, 500)	220
learning_rate	Shrinkage coefficient of each tree	(0.01, 0.2)	0.14
subsample	Subsample ratio of training samples	(0.1, 1)	0.8
max_depth	Maximum depth of a tree	(2, 20)	7

**Table 4 sensors-23-05675-t004:** The *R*^2^ and MAE for training and test datasets (Metrics).

Dataset	MAE	*R* ^2^
Training set	0.053	0.998
Test set	0.381	0.986

**Table 5 sensors-23-05675-t005:** Annual extremes of cable uniform temperatures (*T_u_*).

Year	Temperature *T_u_* (°C)		Year	Temperature *T_u_* (°C)		Year	Temperature *T_u_* (°C)	
	Maximum Temperature	Minimum Temperature		Maximum Temperature	Minimum Temperature		Maximum Temperature	Minimum Temperature
1987	35.84	5.01	1999	35.63	1.91	2011	37.35	4.89
1988	35.68	6.39	2000	36.51	4.25	2012	36.79	3.75
1989	35.03	5.86	2001	37.02	5.99	2013	36.60	5.35
1990	35.14	5.89	2002	38.89	4.27	2014	37.65	4.53
1991	37.18	1.39	2003	37.75	5.45	2015	36.49	5.74
1992	36.06	3.30	2004	37.56	4.59	2016	37.11	0.33
1993	36.04	3.57	2005	37.41	2.75	2017	38.05	7.39
1994	35.41	4.95	2006	37.06	4.45	2018	36.95	3.74
1995	36.39	4.78	2007	36.91	6.41	2019	38.81	8.45
1996	36.92	4.29	2008	36.62	5.54	2020	37.32	5.28
1997	35.13	5.55	2009	36.81	4.69			
1998	36.56	5.53	2010	37.14	2.37			

**Table 6 sensors-23-05675-t006:** Representative values of cable uniform temperatures (*T_u_*).

The Exceeding Probability	2%	1%
The return period	50-year	100-year
Maximum uniform temperature *T_u,max_* (°C)	38.82	39.51
Minimum uniform temperature *T_u,min_* (°C)	1.12	0.27

## Data Availability

The data presented in this study are available on request from the corresponding author.

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
