# Peer review of "Investigation of the Temperature Actions of Bridge Cables Based on Long-Term Measurement and the Gradient Boosted Regression Trees Method"

_sensors, 2023, doi:10.3390/s23125675_

Round 1

Reviewer 1 Report

My comments for the manuscript titled: “Investigation on the Temperature Actions of Bridge Cables Based on Long-Term Measurement and Gradient Boosted Regression Trees Method”

This work studies the thermal distribution, the representative value of temperature, and the time variability of temperatures in the bridge cables. The study has been performed over a year in a practical way on the basis of monitoring the temperature of the cables. The following comments may improve the research on this topic,

*It is mentioned: “The most important results indicated that the temperature distribution is generally uniform along the cross-section without significant temperature gradient”. However, it is questionable cause the type of cross-section may be fruitful to achieve this result.

*The wind and the tension force of the cables may influence the deformation and distribution of temperature for the cables. And the uniform distribution is for a particular situation not an absolute result for the stayed cables.

*In continuation, from this reviewer’s point of view, the cables’ arrangement can also be effectual for the final temperature distribution.

*According to Figure 3, the cable strings are arranged symmetrically. Moreover, the installed and settled sensors are also embedded symmetrically. This may affect the final results.

Reviewer 2 Report

Thank you for this contribution. This is an interesting and timely manuscript. This paper discusses how novel machine learning can be used in earthquake problems on corroded RC columns. The conducted analysis is typically standard and falls within the expected work from such a publication and hence the work merits publication. As such, the authors are invited to properly address the following items:

1. In general, the introduction is light and does not represent state of the art in this domain. The amount of work in this area continues to rise rapidly. For example, the works of Kodur et al. on bridges and fire cover the thermal component of this paper. In addition, the addition of 1-2 pages can help strengthen this section. The following are some review papers on the ML component that can help identify relevant works (https://doi.org/10.1007/s11831-022-09793-w, https://doi.org/10.1016/j.jobe.2020.101827, https://doi.org/10.1016/j.eswa.2020.114060). Evidently, the authors are invited to review the open literature for additional works.

2. The selection of performance metrics in this study is not clear (with MAE and R2). The authors are asked to present their rationale for this metric and not supplement it with other possible metrics. For example, both Alavi et al (2021) and Botchkarev (2019) present a detailed review of the proper selection of performance metrics.

3. It seems that the temperatures are small and remain within the low range (i.e., less than 100C). How do the thermal tests affect the practicality of the cables? 

3. The distribution of data is not clear nor provided by histograms. It is then hard to visualize how the data is distributed. Please provide such plots.

4. For a ML-based work, a question arises as to how our readers can benefit from the developed models. Thus, the authors are advised to consider providing their code and database for interested researchers to extend and benefit from this work. For example, the authors may option to upload this database into Mendeley or attach it to this paper. The same can be done for the code.

Reviewer 3 Report

The Review of the Manuscript entitled: "Investigation on the Temperature Actions of Bridge Cables Based on Long-Term Measurement and Gradient Boosted Regression Trees Method".

The Manuscript is presenting the investigation of the temperature of Bridge Cables based on sensor measurements and gradient boosting technique, which is very important topic in safety analyses of bridge structures. Such cables are crucial parts of structure that are responsible for bearing loads. 

The Structure of the manuscript is relevant for scientific writing and it is also provided in logical manner. The Methodology is sufficiently described that someone else can repeat the study for comparing their results. 

Conclusion is reasonable extension of the results. 

Manuscript is ready for being published after minor revision, following the recommendations:

please pay more attention to describe what is visible in figures (e.g. figure 3) what are the diameters of steel bars?

- please use fitting curves or average values as well while in figure 4 and 5 there is hard to see the results. especially that there are 5 sensors presenting similar results. The comparison is impossible. 

-line 210, the reviewer cant see anything from figure 5 therefore this statement is questionable. 

- figure 18 and figure 19, please revise in terms of fonts and turns of blocks in figure 18.

Round 2

Reviewer 2 Report

.